# Enhanced Removal and Toxicity Decline of Diclofenac by Combining UVA Treatment and Adsorption of Photoproducts to Polyvinylidene Difluoride

**DOI:** 10.3390/polym12102340

**Published:** 2020-10-13

**Authors:** Kristina Fischer, Stephan Sydow, Jan Griebel, Sergej Naumov, Christian Elsner, Isabell Thomas, Amira Abdul Latif, Agnes Schulze

**Affiliations:** Leibniz Institute of Surface Engineering (IOM), Permoserstr.15, D-04318 Leipzig, Germany; kristina.fischer@iom-leipzig.de (K.F.); Stephan.sydow@web.de (S.S.); jan.griebel@iom-leipzig.de (J.G.); sergej.naumov@iom-leipzig.de (S.N.); christian.elsner@iom-leipzig.de (C.E.); isabell.thomas@iom-leipzig.de (I.T.); amira.abdullatif@iom-leipzig.de (A.A.L.)

**Keywords:** phototransformation, diclofenac, PVDF, UVA light, wastewater, wastewater treatment

## Abstract

The occurrence of micropollutants in the environment is an emerging issue. Diclofenac, a non-steroidal anti-inflammatory drug, is one of the most frequently detected pharmaceuticals in the environment worldwide. Diclofenac is transformed by UVA light into different products with higher toxicity. The absorbance of the transformation products overlaps with the absorbance of diclofenac itself and inhibits the ongoing photoreaction. By adding polyvinylidene difluoride (PVDF), the products adsorb to the surface of PVDF. Therefore, phototransformation of diclofenac and total organic carbon (TOC) removal is enhanced and the toxicity decreased. At 15 min and 18 h of UVA treatment, removal of diclofenac and TOC increases from 56% to 65% and 18% to 54%, respectively, when PVDF is present. The toxicity of a UVA treated (18 h) diclofenac solution doubles (from 5 to 10, expressed in toxicity units, TU), while no toxicity was detectable when PVDF is present during UVA treatment (TU = 0). PVDF does not need to be irradiated itself but must be present during photoreaction. The adsorbent can be reused by washing with water or ethanol. Diclofenac (25 mg L^−1^) UVA light irradiation was monitored with high performance liquid chromatography (HPLC), UV-Vis spectroscopy and by analysing the decrease of TOC. The toxicity towards *Vibrio fischeri* was examined according to DIN EN ISO 11348-1: 2009-05. Density functional theory (DFT) was used to simulate the phototransformation products known in literature as well as further products identified via gas chromatography–mass spectrometry (GC-MS). The absorption spectra, reaction enthalpies (ΔH) and Gibbs free energy of reactions (ΔG) were calculated. The combination of UVA irradiation of diclofenac with adsorption of photoproducts to PVDF is unique and opens up new possibilities to enhance removal of pollutants from water.

## 1. Introduction

Diclofenac is one of the most frequently detected pharmaceuticals in the environment worldwide [1]. It is a nonsteroidal anti-inflammatory drug, which prevents inflammation by inhibiting prostaglandin synthesis and is used to reduce pain and to treat inflammatory diseases [2]. After consumption, diclofenac is excreted in a metabolized and parental form [3]. In Germany some 85 tons of diclofenac are disposed of per year [4]. The release of diclofenac into the environment leads to a high ecotoxicological risk [5,6] and phototoxicity [7]. The predicted no-effect concentration for diclofenac was set to 0.1 μg L^−1^ by the European Commission in 2015 [8]. In Germany the yearly average concentration of diclofenac exceeded the defined predicted no-effect concentration with values up to 1 μg L^−1^ in 2015 at 21 out of 24 monitoring stations [4]. In 37 countries (out of 50) the reported diclofenac concentrations found in water exceed the predicted no-effect concentration of 0.1 μg L^−1^ [1]. Diclofenac evidently shows toxic effects at these concentrations present in the surface water [5,6] and was the cause of a mass extinction of vultures in India and Pakistan [9,10].

The main route of diclofenac into the environment is through wastewater treatment plants (WWTPs) [11]. Elimination of diclofenac in WWTPs is insufficient, as they were not designed to eliminate pharmaceuticals like diclofenac from water. The estimated removal efficacy for diclofenac using conventional WWTPs vary from 39% [12,13] over 43% [14] to 70% [15]. In activated sludge only 30 to 35% of diclofenac is removed [16]. The current state to cope with diclofenac in WWTPs is to use an additional treatment stage. Since 2016 Switzerland regulates the concentration of certain micropollutants in water [17], thus scientists look closely at Switzerland’s strategy and experience. Two main treatment methods are used in Switzerland [17]: ozonation [18] and activated carbon [19]. Both processes are accompanied by high operating expenses like pre-treatment and post-treatment [20,21,22]. Advanced oxidation processes (AOPs, like photocatalysis, photo-Fenton, ozonation or electrochemical oxidation) are highly discussed and investigated processes to eliminate micropollutants from water [23,24,25,26]. Reactive oxygen species (hydroxyl radicals, hydrogen peroxide, ozone, and superoxide anion radicals) are generated and completely mineralize micropollutants with a low selectivity to carbon dioxide, water, and inorganic ions or acids. The main drawbacks of photocatalysis are the high band energy of the semiconductor material, which are mostly not visible light active and show poor activity, agglomeration of used nanoparticles, the difficulty to separate the catalyst after photocatalysis from the aqueous media and the reusability of the catalyst [27]. Ozonation or degradation of pollutants via hydrogen peroxide are always accompanied by the addition of toxic and expensive oxidants. Also, AOPs generate several oxidation by-products with a high toxic potential [28]. Conventional treatment methods have many problems and drawbacks. Therefore, further studies to evaluate new strategies to remove pollutants like diclofenac from water are highly needed.

In the environment, diclofenac is degraded by sunlight in two different pathways. One way is the direct photolysis caused by absorption of solar radiation. The other way is the indirect degradation caused by strong oxidizing species like hydroxyl radicals or singlet oxygen formed due to solar radiation [29,30,31]. The reduction of diclofenac under irradiation is influenced by the absorption cross section of the UV light and the quantum yield [32]. The UV absorption cross section is expressed as a molar extinction coefficient, which measures how strongly a chemical specie absorbs light at a certain wavelength [33]. Further factors affecting the kinetic of a photochemical reaction are the intensity and wavelength of the incident radiation, the optical path of the radiation, the nature of the irradiated compound, the concentration and the solution in which it is present [34]. Diclofenac can be excited by wavelengths up to 320 nm. The terrestrial solar UV radiation starts at 295 nm [35].

Excitation of diclofenac with UVA light (>300 nm) leads to different reaction pathways like ring closure, decarboxylation [36], hydrolysis [37] or dehydration [36,37], thus a multitude of different phototransformation products (like carbazoles [36], quinones [38,39], dimers [39,40]) are formed. The transformation of diclofenac by UV light is highly inhibited by the formation of products. These products are able to absorb the incident radiation in the same wavelength range as diclofenac and with higher extinction coefficients [39]. The excitation of diclofenac is delayed as the phototransformation products absorb the incident radiation. The more diclofenac gets converted into inhibiting products the less radiation is available for the conversion of diclofenac [37,41]. This transformation rate reduction of diclofenac over time by the above described reasons was described in the literature as an “inner” filter effect [40]. The removal of phototransformation products during UVA light irradiation of diclofenac would increase the reaction rate, as transformation would not be hindered by these products anymore. The “inner” filter effect is reduced.

In this work, an adsorbing material (hydrophobic polyvinylidene difluoride, PVDF) is introduced during UVA irradiation of diclofenac. Up to date PVDF was only applied as a support for photocatalysts [42,43], or molecularly imprinted polymer nanoparticles [44] which remove diclofenac from water. Here, PVDF adsorbs phototransformation products but not diclofenac itself [44]. The inhibiting products are drawn from the solution to reduce the “inner” filter effect and increase UVA transformation of diclofenac. So far, it is the first time, that UVA irradiation and adsorption of phototransformation products are combined to show an increased removal of a pollutant from water. This work demonstrates a new concept, which opens up new possibilities for pollutant elimination from water.

Experiments were performed under an UVA lamp with and without PVDF. The diclofenac transformation was monitored with high performance liquid chromatography (HPLC), UV-Vis spectroscopy and by analysing the decrease of total organic carbon (TOC). The toxicity towards *Vibrio fischeri* was examined according to DIN EN ISO 11348-1: 2009-05. Density functional theory (DFT) was used to simulate the phototransformation products known in literature [36,39] as well as further products identified via gas chromatography–mass spectrometry (GC-MS). The absorption spectra, reaction enthalpies (ΔH) and Gibbs free energy of reactions (ΔG) were calculated. Reusability and recovery tests of PVDF were performed under the best-established conditions.

## 2. Materials and Methods

### 2.1. Materials

Distilled water was received from Milli-Q Quantum Tex (Merck KGaA, Darmstadt, Germany, 18.2 MΩ cm, 3 ppb total organic carbon). Diclofenac sodium salt was obtained from Cayman Chemical Company (Ann Arbor, MI, USA). Ethanol (≥96%, denatured) and PVDF flat-sheet membranes with a pore size of 0.45 µm were acquired from Carl Roth GmbH + Co.KG (Karlsruhe, Germany). Soda-lime petri (Steriplan, Duran Group) dishes were purchased from DWK Life Sciences GmbH (Wertheim, Germany).

### 2.2. Density Functional Theory Calculations

Density functional theory (DFT) calculations were carried out systematically using PBE0-D3 density functional [45,46,47] as implemented in the program Jaguar program package [48]. PBE0-D3 functional includes London dispersion interactions [49]. The molecular geometries and energies of the completely simulated molecules were calculated using 6-31G (d,p) basis set. Frequency calculations were done at the same level of theory to characterize the stationary points on the potential surface and to obtain the total enthalpy (H_tot_) and Gibbs free energy (G) at a standard temperature of 293 K using unscaled vibrations. To take the solvent effect of water on the structure and reaction parameters of studied molecules into account, the calculations were done using Jaguar’s dielectric continuum Poisson-Boltzmann solver (PBF) [50]. The electronic transition spectra were calculated in a solvent with the time-dependent (TD) [51] TDDFT PBE0-D3/6-31G(d,p)/PBF method.

### 2.3. Irradiation Experiments

The PVDF membranes were cut into 47 mm diameter discs, or half discs with a diameter of 77 mm for tests with shielded and reused PVDF, respectively. Prior to irradiation experiments, the hydrophobic PVDF adsorbent was pre-wetted with ethanol for 2 min followed by washing with water for 2 and 5 min. Further, the PVDF adsorbent was placed in 78 mm soda-lime Petri dishes. 25 mL of diclofenac sodium salt solution (25 mg L^−1^) in deionized water (Milli-Q water filtration station, Millipore, Burlington, MA, USA) was added to petri dishes with and without PVDF. An 83 mm Petri dish was placed as a cover. The Petri dish was placed on a radial shaker. The sample was irradiated with UVA light (combined radiant flux density of 17 mW cm^−2^, Heraeus Original Hanau Suncare OH M 21/25 R Slim 25 W Tanning Tube from HN Sunlight GmbH, Hanau, Germany) for up to 18 h. Samples were taken afterwards to monitor the transformation process via UV-Vis spectroscopy, HPLC, TOC and GC-MS.

For experiments with shielded (not irradiated) PVDF or reusability tests, the half discs were placed in an in-house manufactured 78 mm petri dish (see Figure 1). Glass mounts hold the PVDF discs in place. The covering petri-dish was half-shielded with an opaque reflective foil (see Figure 1) and placed in a way above the PVDF discs, so that the PVDF adsorbent was either shielded or irradiated.

PVDF was also tested for re-use up to five times. The irradiation time was set to 30 min. In between the irradiation, the PVDF adsorbent was either directly reused, washed 2 times with water (each 25 mL) for 2 and 5 min or washed in ethanol (25 mL) for 2 min and twice in water (each 25 mL) for 2 and 5 min.

To gain knowledge about the effect of adsorption of transformation products during photoreaction and afterwards a PVDF disc was added for 1 h after 18 h of UVA irradiation (without PVDF).

### 2.4. Characterization of Treated Diclofenac Solution and Extracts

The standard deviation of the data was the basic statistical tool to give the amount of variation of the stated values. The confidence interval was estimated for the TU values (toxicity).

UV-Vis spectroscopy was performed at a Tecan Infinite^®^ 200 microplate reader (Tecan Trading AG, Männedorf, Switzerland). Samples of 60 μL were measured (triplicate) in a UV-STAR^®^Plate, 384 Well, F-Bottom, μClear^®^, clear (Greiner Bio-One GmbH, Frickenhausen, Germany) well plate from 230–315 nm (5 nm slit width) and 316–400 nm (9 nm slit width) in 2 nm steps.

Reversed-phase HPLC measurements were conducted at a Dionex UltiMate 3000 HPLC system equipped with pump, autosampler and a variable wavelength detector (Thermo Fisher Scientific, Waltham, MA USA) with a Kinetex 5 μm Biphenyl 100 A Column 250 × 4.6 mm (Phenomenex, Torrance, CA, USA). The analysis was performed with Dionex™ Chromeleon™, Version 6.80 SR14 (Thermo Fisher Scientific). The measurement was executed for 30 min with 60% acetonitrile and 40% of 0.5 mol-% trifluoroacetic acid at isocratic conditions. The detection wavelength was set at 278 nm.

The total organic carbon (TOC) was quantified (triplicate) at a liquiTOC II (Elementar, Langenselbold, Germany) with high temperature oxidation between 850–900 °C and a wide range NDIR photometer. The concentration range was set to 20 ppm and calibration was performed prior to the tests. Samples were collected in glass vials at a volume of 25 mL and covered with tin foil.

The samples were lyophilized prior to GC-MS measurements. The transformation products adsorbed on PVDF were extracted in a 10 mL ethanol and water solution (1:1). After 2 min of shaking, the PVDF adsorbent was removed. The extract as well as the irradiated diclofenac solutions were lyophilized. The dry residues after lyophilization were suspended in 1 mL of 10% ethanol solution. The analysis was carried out on a 6890N Network GC System (Agilent Technologies, Santa Clara, CA, USA) connected to an Agilent 5973 Network mass selective detector. The measurements were performed with the help of a Gerstel MultiPurposeSampler (Mülheim an der Ruhr, Germany). An incubation time of 20 min with a SPME fiber (65 µm PDMS/DVB, Supelco, Bellefonte, PA, USA) at 303.15 K was chosen to reduce potential thermal decomposition. The analyst was separated on an Agilent HP-5MS capillary column (60 m, 0.25 mm ID, 0.25 µm df) with helium as carrier gas (1.4 mL min^−1^ constant flow). The GC program started at 323.15 K and was increased to 523.15 K at a rate of 4 K min^−1^. The final temperature was held for 20 min. Electron impact ionization was performed at 70 eV. A full scan mode over a mass range of m/z = 20–800 was performed.

The toxicity tests on the marine luminescent bacteria *Vibrio fischeri* were performed according to DIN EN ISO 11348-1: 2009-05 by the Umweltanalytische Labor of the Institut für Siedlungswasserwirtschaft of RWTH Aachen with a LUMIStox using LUMIStherm System (Hach Lange GmbH, Düsseldorf, Germany). The test consisted of seven dilutions. The pH of the samples was corrected to 6.0 (*SD* = 0.2) before toxicity assessment tests using 0.1 M NaOH. According to the applied procedure, the self-produced (freshly grown) bacteria were reconstituted with reactivation solution, to provide a stock suspension of test organisms which was maintained at 15 °C and used to perform the test. The measurement of the bioluminescence was carried out at the start and after 30 min and the luminescence of the dilutions was compared with the negative control. A positive control was also included. Each dilution was tested in parallel. The data were automatically evaluated with the program LUMISsoft (Hach Lange GmbH) and the statistics program ToxRat (ToxRat Solutions GmbH, Alsdorf, Germany). The toxicity was expressed as toxicity units (TU) and G_L_ levels. The TU was calculated by dividing 100 by the effective concentration causing 50% reduction of bioluminescence (TU = 100/EC_50_%). The G_L_ value is the dilution level at which the inhibition is below 20% for the first time. All values below 20% inhibition are normal test fluctuations. Level 1 (80% volume fraction), level 4 (25% volume fraction), level 6 (16.66% volume fraction), level 8 (12.5% volume fraction).

## 3. Results and Discussion

### 3.1. Formation of Phototransformation Products and Their Absorption

Phototransformation products [36,38,39,40] are formed during the irradiation of diclofenac with UVA light. The Gibbs free energy G and the total enthalpy H_tot_ were simulated via PBE0-D3 method for phototransformation products identified via GC-MS (see Figure 2). The products II and VIIIa/b were not found via GC-MS, but are known from literature [36,39] and therefore added for completion. The reaction enthalpy (ΔH, difference of the total enthalpies H_tot_ of reactants and products) and Gibbs free energy (ΔG, difference of the Gibbs free energies G between the reactants and products) of possible reactions during irradiation of diclofenac have been calculated and are shown in Figure 2.

Diclofenac itself is stable without irradiation. All primary reactions taking place during irradiation of diclofenac, except for reaction (5), occur after an intersystem crossing and excitation into the triplet state (T1). The excited T1 state is likely to react in multiple pathways. Reaction (1) eliminates HCl leading to product II. A radical pathway is considered (1b). This process is endergonic and endothermic; thus, it will not take place. The 6π-electro cyclisation (1a), as suggested by Encinas et al. [36], is the dominant pathway. Further, dechlorination from an excimer of II (T1) and II (S0) to radical intermediates is reported [36]. A decarboxylation is also possible. According to the pH-value during irradiation of diclofenac, the decarboxylation of I takes place in two different ways and depends on the protonation of the carboxyl group function (2a) or (2b). For both schemes the radical intermediate (IIIa) is proposed. Product VII is formed by combining HCl-elimination (1) and decarboxylation (2a/b). An ongoing reaction for product II (6a) is more likely to occur than for product III (6b), but both are possible at the given conditions. Pathway (3) is a dehydration reaction resulting in ring-closure and forms the amide IV. A hydrolysis of diclofenac is also possible. The amine function is retained on one of the cleavage products. A slight energetic tendency for reaction (4b) is present leading to Vc and Vd compared to the other cleavage products Va and Vb. As Keen et al. [39] reported diclofenac or some of its transformation products are able to photosensitize singlet oxygen leading to a dimerization. A reaction with singlet oxygen (5) forms the diclofenac epoxide VI. Molecule IX is either a product from epoxidation of the dehydrated product IV (8a) or a dehydration of the diclofenac epoxide VI. Both reactions are not favored under the given reaction parameters. The dimerization of molecule VI leads to the direct product VIIIa (7a) or to VIIIb (7b) after oxygen elimination. The dimers VIIIa and VIIIb are expected to react further on as all functional groups are still present and not all reactions described for diclofenac, except for the epoxidation, should be inhibited.

The absorption spectra of above discussed transformation products have been simulated. A selection is shown in Figure 3a. The absorbance of diclofenac overlaps with the absorbance of phototransformation products. In the irradiated region, product IV shows a neglectable absorbance, while all other products have a higher absorbance than diclofenac. The dimers (VIIIa and VIIIb) have a higher molar absorptivity than diclofenac [39]. The transformation of diclofenac via UVA light is therefore inhibited. A competition between the phototransformation product itself and diclofenac to absorb the UVA light is the result. This is described as an “inner” filter effect by Agüera et al. [40]. Transformation of diclofenac will be decelerated. The increase of absorbance due to irradiation with UVA light is clearly observed in Figure 3b. With increasing irradiation time, the absorbance enhances at wavelengths between 230–270 nm and 300–400 nm. The formation of transformation products as discussed before is the reason for this increase. On the other side, the absorbance of diclofenac decreases between 270–300 nm. As diclofenac is degraded and transformed to products, the absorbance decreases while it increases at wavelengths of transformation products.

### 3.2. Transformation of Diclofenac via UVA Light in the Presence of PVDF

Diclofenac is transformed in the presence of UVA light (see Figure 4). Within a time of 90 min, diclofenac is eliminated from the tested solution (Figure 4a). The total organic carbon (TOC) is removed to a lower degree (Figure 4b). After 18 h of irradiation with UVA light, only 17.8% (*SD* = 2.7%) of TOC is mineralized. Overall, phototransformation depends on many parameters like molar absorption coefficient, quantum yield, intensity of emitted irradiation (incident photon flux), radiation wavelength and photoreactor design. Thus, direct comparison with literature values is difficult. Diclofenac (30 and 10 mg L^−1^) is transformed by UVC irradiation (254 nm) within 9 min [52] and 6 min [53], respectively. Complete phototransformation of diclofenac (30 mg L^−1^) via UVA (365 nm) is reported to take 29 h [52]. UVA irradiation of diclofenac (0.5 mg L^−1^) with similar wavelengths (313–578 nm) results in lower transformation of 20% after 60 min [28] compared to over 90% in this study. Diclofenac is excited by light up to wavelengths of 320 nm with an absorption maximum at 278 nm (see Figure 3b). Thus, the difference in phototransformation kinetics in literature to the here reached values is due to (amongst others) the applied irradiation wavelength. The TOC removal after complete diclofenac transformation is higher in this study compared to literature value of < 10% for UVA and UVC irradiation [52].

Adding PVDF increases the reaction rate of diclofenac transformation and TOC removal. After 20 min of transformation, only 64.9% (*SD* = 0.2%) of diclofenac is transformed without, compared to 72.3% (*SD* = 0.2%) of diclofenac elimination with PVDF present during UVA irradiation. The adsorption of diclofenac itself at PVDF is neglectable [44] with 4% *(SD* = 0.4%). The TOC is also removed to a higher value when PVDF is present during UVA irradiation. Again, adsorption of diclofenac itself is insignificant. The removal of TOC increases from 17.8% (*SD* = 2.7%) to 54.0% (*SD* = 1.3%) when PVDF is present during 18 h of UVA light irradiation. Adding PVDF after 18 h of photoreaction to the irradiated solution leads to a removal of 37.5% (*SD* = 2.4%) of TOC. TOC removal via UVA irradiation is highly increased with PVDF present. Even strong UVC irradiation by Kovacic et al. [52] resulted in lower TOC removal of 10% after 100 min of irradiation compared to values of 26% and 40% after 60 min and 120 min in this study, respectively.

#### 3.2.1. Effect of PVDF

In Section 3.1, the generation of inhibiting phototransformation products has been described. The phototransformation products show a high absorption, thus are hindering the transformation of diclofenac. The increase of absorbance during irradiation is shown in Figure 3. By adding PVDF the absorbance increase is reduced, thus higher diclofenac transformation is observed (Figure 4a and Figure 5a). In Figure 5b, the change of total absorbance (area of absorbance at non-irradiated and irradiated wavelengths) in percent over time of diclofenac transformation is presented. A steep increase of absorbance at irradiated wavelengths (309–400 nm) in the first 15 to 20 min is observed for diclofenac irradiated with and without PVDF. After 20 min of diclofenac irradiation, the steep increase in absorbance diminishes slowly. Adding PVDF during diclofenac irradiation with UVA light leads to a faster diminishment with a marginal increase in absorbance after 30 min. PVDF acts as an adsorbent of phototransformation products, thus the absorbance increase is reduced and the actual transformation of diclofenac via UVA light is increased. The absorbance at non-irradiated wavelengths between 230–309 nm is also increasing until 15 min of diclofenac irradiation. However, the increase is very low, as the absorbance of diclofenac is also decreasing in that wavelength range due to transformation of diclofenac. After 15 min, the absorbance of the irradiated diclofenac solution with PVDF even decreases due to adsorption of products at PVDF and increased transformation of diclofenac via UVA light.

The behavior of a transformation product during irradiation of diclofenac with UVA light, with and without PVDF was further investigated. After 18 h of irradiation without PVDF the product X (see Figure 2 in Section 3.1. for structural formula; m/z 231) was found with a proportion of 96% regarding all detected molecules via GC-MS (see Figure 6a). Only 38% of the phototransformation product X was found when PVDF was present during irradiation with UVA light. The extract of PVDF (which was present during 18 h of irradiation of diclofenac with UVA light) was examined via GC-MS. The phototransformation product X was found in the extract and made up over 50% next to other products like III, IV and Vb. Thus, it can be assumed that phototransformation product X adsorbs to PVDF during diclofenac irradiation with UVA light.

The transformation of diclofenac with UVA light with a shielded (not irradiated, as shown in Figure 1) but present PVDF is displayed in Figure 6b. The degree of transformation is the same compared to non-shielded PVDF. After 30 min of irradiation, 62.9% (*SD* = 1.1%) and 63.8% (*SD* = 1.1%) of diclofenac is phototransformed with UVA light with PVDF irradiated and shielded, respectively. Thus, even if PVDF is not irradiated itself, it accelerates transformation of diclofenac. Hence, PVDF itself does not need to be irradiated in order to increase transformation of diclofenac with UVA light. PVDF acts only as an adsorbent for the phototransformation products and does not directly transform diclofenac and its products nor generates active species (like radicals) which could transform diclofenac, too. PVDF just needs to be present in the solution during irradiation.

Adding PVDF after irradiation is completed (18 h UVA treatment without PVDF) increases the amount of TOC removed from the solution from 17.9% (*SD* = 2.7%) to 37.5% (*SD* = 2.4%) (see Figure 4b). Transformation products adsorb at the PVDF surface and lead to an enhanced TOC removal. However, if PVDF is present during 18 h of UVA treatment the removal is as high as 52.3% (*SD* = 1.5%). The adsorption of transformation products during irradiation enhances the photoreaction of diclofenac and its transformation products.

It can be concluded that phototransformation products adsorb to PVDF during photoreaction. Therefore, the transformation of diclofenac is faster as phototransformation products are actively removed from the solution during UVA irradiation.

#### 3.2.2. Regeneration of PVDF

As discussed in previous sections PVDF acts as an adsorbent for phototransformation products of the phototransformation of diclofenac. Every adsorbing material has a limited capacity and will stop adsorbing after collecting a certain number of molecules. Therefore, PVDF was reused up to five times for phototransformation of diclofenac (0.5 h) and the alteration of diclofenac transformation was analyzed after each usage (see Figure 7). The alteration of diclofenac transformation after each cycle was set in relation to the transformation after the first UVA irradiation. Reusing PVDF without washing leads to a reduction in diclofenac transformation since less phototransformation products can adsorb to PVDF. After the second usage, the transformation of diclofenac decreased by 10.6% (*SD* = 1.8%) with further reduction to 15.6% (*SD* = 3.5%) after the fourth usage. The removal of TOC after repetitive usage for up to five times decreases from 3.9% (*SD* = 0.7%) to 8.5 (*SD* = 1.1%), respectively. However, PVDF can be regenerated. The adsorbing phototransformation products can be desorbed by water (partially) or ethanol (see Figure 7). The alteration of transformation of diclofenac after 0.5 h of UVA irradiation with repeatedly used and regenerated PVDF does not change significantly regarding the different irradiation cycles. Washing PVDF in between UVA irradiation with ethanol regenerates the adsorbent totally. The same amount of diclofenac is transformed, and equal amount of TOC is removed from the solution when PVDF is regenerated with ethanol in between. Water as a washing agent partially regenerates PVDF. A decrease in transformation of diclofenac by 4.9% (*SD* = 0.4) after the fifth cycle is observed, while the reduction of TOC stays constant.

Thus, the PVDF adsorbing capacity can be maintained by simply washing the membrane in between with water or ethanol.

### 3.3. Toxicity Analysis

Diclofenac is classified by the EU Directive 93/67/EEC as “harmful to aquatic organisms” [54,55] and due to its low log K_ow_ of 4.51 as toxic [56]. The EC_50_ toxicity towards *Vibrio fischeri* was determined to be 19.7 mg L^−1^ (with a confidence interval of 17.9 to 22.2) which is comparable to literature value (15.78 mg L^−1^ [56]).

As it can be seen in Figure 8, the toxicity towards *Vibrio fischeri* increases with ongoing phototransformation until 90 min of irradiation. Diclofenac completely transforms within the first 90 min of UVA irradiation. The phototransformation products are more toxic towards *Vibrio fischeri* than diclofenac itself. The toxicity unit (TU = 100/EC_50%_, EC_50%_ is the effect concentration at which 50% of effect occurs; the higher the TU, the higher the toxicity) and G_L_ level (dilution level at which the inhibition is below 20% for the first time; the higher the G_L_ level, the higher the toxicity) at 90 min of UVA irradiation doubles compared to an untreated diclofenac solution. The evaluation of the toxicity of diclofenac treated with UVA light is controversial discussed [52,56,57,58]. But the conditions (concentration of diclofenac, UVA light wavelength, UVA light intensities, additives in water, tested biological organism) do differ and lead to different transformation products and concentrations of these products. Hanamoto et al. [57] and Kovacic et al. [52] identify a decrease of toxicity towards *Vibrio fischeri* after sunlight and UVA (365 nm) light treatment, respectively. Contrary, Schmitt-Jansen [58] studies showed an increase of toxicity towards the algal *S. vacuolatus* after day light treatment. A carbazole-type compound and diphenylamine derivates, which are generated during phototransformation of diclofenac, were identified to be the toxic transformation products.

The toxicity increases regarding TU (50% effect) for the phototransformation during the first 90 min with and without PVDF is similar. The toxicity shift regarding G_L_ levels (20% inhibition) is slightly different analyzing diclofenac irradiation with and without PVDF. After 30 min of UVA irradiation with PVDF the G_L_ is two levels higher, thus more toxic and only 16.66% of volume fraction is needed to gain a 20% inhibition compared to 25% of volume fraction without PVDF. After 90 min of photoreaction the G_L_ level is lower for the solution with PVDF compared to without PVDF (G_L_ level 8 to 6, respectively), thus the toxicity is reduced with PVDF. After 18 h of photoreaction with PVDF the toxicity declines. An EC_50%_ value could not be evaluated, and the G_L_ level is 1. The 18 h UVA light treated diclofenac solution without PVDF shows towards *Vibrio fischeri* a higher toxicity (regarding TU and G_L_) than the untreated diclofenac solution. Phototransformation product X (see Figure 2 in Section 3.1. for the structural formula), which is a diphenylamine derivative [58], might be one of the toxic products. After 18 h of irradiation without PVDF, product X (see Figure 6a in Section 3.2.1.) was found with a proportion of 96% regarding all detected molecules via GC-MS. Only 38% of the phototransformation product X was found when PVDF was present during irradiation with UVA light. The amount of product X in the solution after UVA light treatment with PVDF is reduced, thus the toxicity decrease might be correlated to its enhanced removal.

The treatment of diclofenac with UVA light leads to an increase of toxicity regardless of the presence of PVDF within the first 90 min. After 18 h of irradiation the diclofenac solution with PVDF shows almost no toxic effect towards *Vibrio fischeri* anymore, while the toxicity is doubled without PVDF. The addition of PVDF during diclofenac UVA irradiation leads to a reduction of toxicity.

## 4. Conclusions

Diclofenac is transformed to different products due to irradiation with UVA light. The absorbance of diclofenac overlaps with the absorbance of phototransformation products, thus transformation is inhibited. Transformation of diclofenac can be increased by adsorbing phototransformation products.

The addition of PVDF during UVA irradiation increased the transformation of diclofenac from 64.9% (*SD* = 0.2%) to 72.3% (*SD* = 0.2%) (after 20 min of UVA irradiation), enhanced TOC removal from 17.9% (*SD* = 2.7%) to 52.3% (*SD* = 1.5%) (after 18 h of treatment) and decreases the toxicity towards *Vibrio fischeri* from a TU value of 10.2 to 0 (after 18 h of treatment). TOC removal of diclofenac via UV irradiation without PVDF is found to be low in the literature (<10%), but can be raised by the addition of PVDF to over 26%. The PVDF does not need to be irradiated itself during reaction as it only acts as an adsorbing material. The adsorbing capacity can be regenerated by washing PVDF with water or ethanol. As UVA degradation of diclofenac is incomplete, toxicity increases. The addition of PVDF enhances the removal and toxic phototransformation products are removed from the solution.

This concept of increasing transformation by adsorbing phototransformation products which inhibit the reaction could be enhanced with other adsorbing polymers (e.g., PES, PTFE, PE) having different properties (e.g., surface charge, hydrophilicity, functional groups). As the polymer does not have to be irradiated itself, many different materials can be applied. The concept could be also adopted to other UVA phototransformable pollutants like sulfamethoxazole or ketoprofen. The development of a continuous reactor is required to scale up the process. Further studies will be undertaken to study diclofenac removal in a continuous reactor where diclofenac in water is flowing over PVDF with simultaneous UVA irradiation. Different volume, concentration, flow velocities and surface area will be changed and tested.

Combining UVA irradiation and adsorption of phototransformation products to a polymer could improve the elimination of many other pollutants from water and decrease its toxicity. Toxic phototransformation products which are generated during UVA irradiation of pollutants are removed via adsorption. The concept is unique, improving different parameters like adsorbing polymer, UVA irradiation and reactor design, could lead to a promising technique to remove UVA phototransformable pollutants and its transformation products from water.

## Figures and Tables

**Figure 1 polymers-12-02340-f001:**
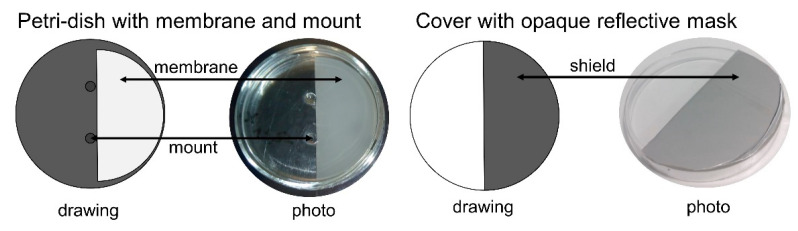
Experimental setup for experiments with shielded PVDF. The adsorbent was placed in the petri-dish and is held in place by glass mounts. A cover that is half-coated with an opaque reflective foil was placed in a way that the adsorbent was either shielded or irradiated.

**Figure 2 polymers-12-02340-f002:**
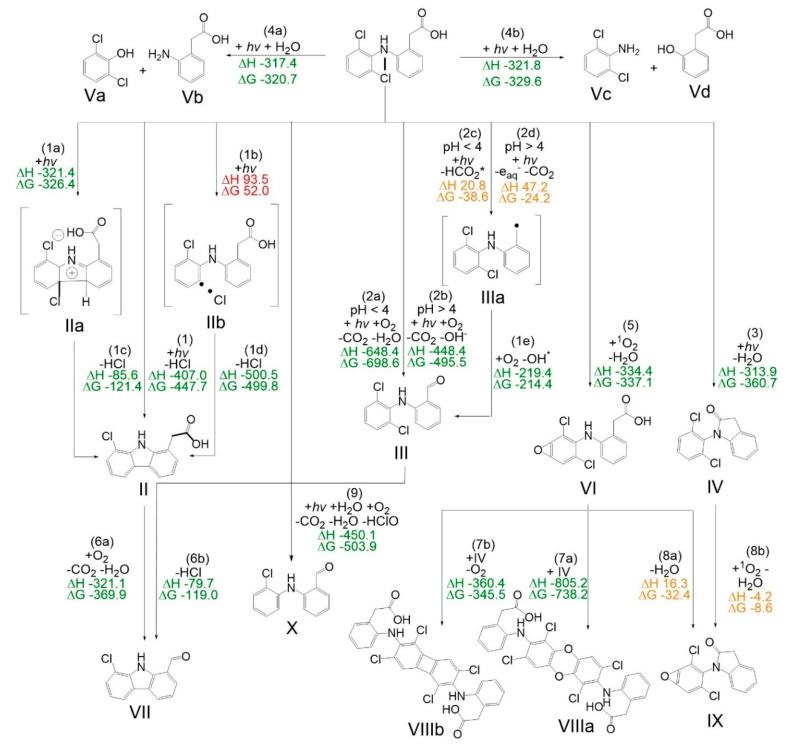
Suggested reaction pathway of the irradiation of diclofenac with UVA light with added reaction enthalpies (ΔH) and the Gibbs free energy of the reactions (ΔG) as calculated by DFT. The reaction energies are colored corresponding to their probability of occurring under the given reaction conditions. Green reactions show a high likelihood, orange reactions can take place and red reactions are not feasible within the given conditions.

**Figure 3 polymers-12-02340-f003:**
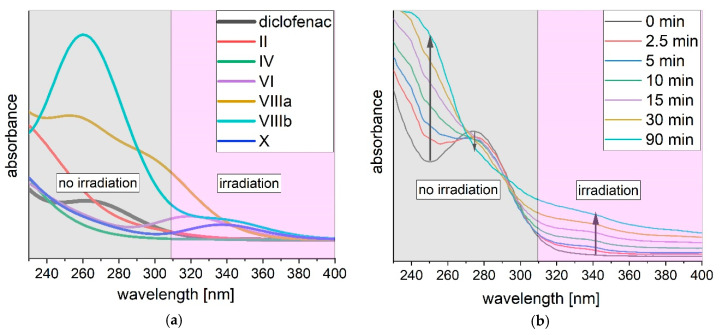
(**a**) Simulated absorption spectrum of diclofenac and selected transformation products (II–X). (**b**) Measured absorption spectrum of diclofenac at different irradiation times. The UVA lamp light emission ranges from 309–400 nm.

**Figure 4 polymers-12-02340-f004:**
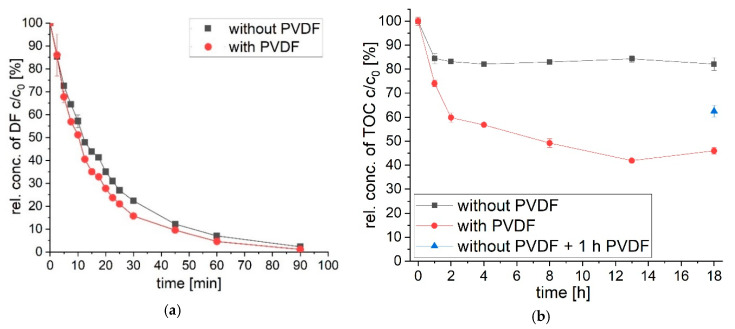
The transformation of diclofenac with UVA light over time with and without a PVDF is followed by analyzing the concentration decrease of diclofenac via HPLC (**a**) and by investigating the removal of organic carbon via TOC analysis (**b**). A PVDF membrane was added for 1 h (no UVA) after 18 h of UVA treatment of a diclofenac solution (blue triangle in (**b**)).

**Figure 5 polymers-12-02340-f005:**
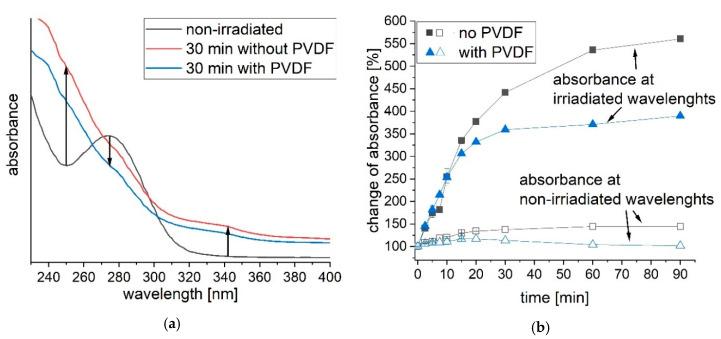
(**a**) Absorbance spectrum of non-irradiated and 30 min UVA irradiated diclofenac with and without PVDF. (**b**) The change of absorbance is shown over time for diclofenac UVA irradiation with and without PVDF. The area under the absorbance vs. wavelength was calculated for the non-irradiated region from 230–309 nm and the irradiated region from 309–400 nm. The change of absorbance was calculated by setting the area of absorbance of the non-irradiated diclofenac solution at 100%.

**Figure 6 polymers-12-02340-f006:**
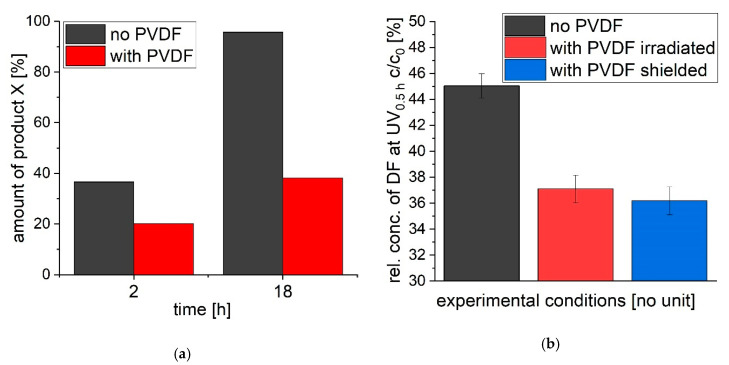
(**a**) Amount of product X in relation to other phototransformation products found via GC-MS after 2 h and 18 h of irradiation with and without PVDF. (**b**) The transformation of diclofenac with UVA for 0.5 h followed by analyzing the relative concentration of diclofenac via HPLC is shown. The irradiation was performed without PVDF, with PVDF irradiated during transformation and with PVDF shielded (not irradiated) during photoreaction.

**Figure 7 polymers-12-02340-f007:**
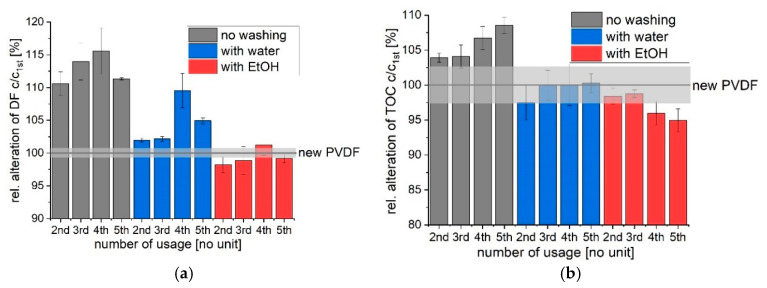
PVDF was reused up to five times (each 0.5 h UVA light irradiation) and either washed with water or ethanol in between or used untreated. Transformation with fresh and unused PVDF was set to 100% and is indicated as new PVDF. The relative alteration of transformation of diclofenac (DF) in (**a**) and TOC in (**b**) in relation to unused PVDF is shown.

**Figure 8 polymers-12-02340-f008:**
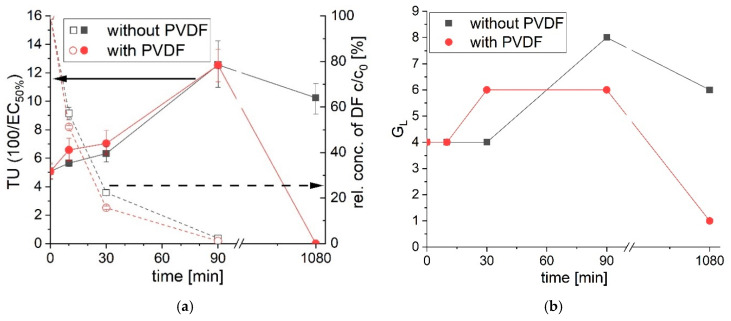
(**a**) Change of toxicity to *Vibrio fischeri* after irradiation of diclofenac with UVA light with and without PVDF (left axis) expressed in toxicity units (TU = 100/EC_50%_, EC_50%_ is the effect concentration at which 50% of effect occurs; the higher the TU, the higher the toxicity). The concentration decrease of diclofenac after irradiation was analyzed via HPLC (right axis). In (**b**) is the toxicity towards *Vibrio fischeri* after irradiation of diclofenac with UVA light expressed in G_L_ levels. The G_L_ value is the dilution level at which the inhibition is below 20% for the first time (the higher the G_L_ level, the higher the toxicity). All values below 20% inhibition are normal test fluctuations. Level 1 (80% volume fraction), level 4 (25% volume fraction), level 6 (16.66% volume fraction), level 8 (12.5% volume fraction).

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
