# Peer review of "Enhanced Removal and Toxicity Decline of Diclofenac by Combining UVA Treatment and Adsorption of Photoproducts to Polyvinylidene Difluoride"

_polymers, 2020, doi:10.3390/polym12102340_

Round 1
Reviewer 1 Report
This study investigated the photodegradation of diclofenac with the presence of PVDF membrane. The results showed that PVDF provided a new method for toxic intermediates pollutants removal. In my opinion, the work is novel and well-written. After minor revisions, this paper can be published in this journal. The following is the specific comments for this paper.

- In the introduction section, kindly indicate some situations that PVDF membrane are widely used with DCF removal.
- The adsorption kinetics of PVDF membrane should be indicated.
- The conclusion section should be more concise.
Author Response
Dear Editior,
We appreciate the thorough examination of the manuscript by the reviewers, and we carefully revised the manuscript for obtaining now a better quality of the paper. The changes have been made using the "Track Changes" function in Microsoft Word. Therefore, I would like to resubmit our manuscript and hope we could eliminate all mistakes and misunderstandings according to the reviewer’s suggestions.
For easier review, we directly answered the reviewers comments (see below). Finally, I would like to thank the reviewers for the very helpful comments and work he/she put in our manuscript!
With best regards,
Agnes Schulze
Review 1:
This study investigated the photodegradation of diclofenac with the presence of PVDF membrane. The results showed that PVDF provided a new method for toxic intermediates pollutants removal. In my opinion, the work is novel and well-written. After minor revisions, this paper can be published in this journal. The following is the specific comments for this paper.
Comment 1: In the introduction section, kindly indicate some situations that PVDF membrane are widely used with DCF removal.
In the introduction section information has been added where PVDF is used to remove PVDF.
Line 96-98: Up to date PVDF was only applied as a support for photocatalysts [42,43], or MIPs (molecularly imprinted polymers nanoparticles) [44] which removed diclofenac from water.
Comment 2: The adsorption kinetics of PVDF membrane should be indicated.
Adsorption kinetics cannot be calculated as UVA phototransformation of diclofenac is taken place at the same time of adsorption of these products to PVDF. To calculate adsorption kinetics the amount of products in solution which adsorb on PVDF has to be known and should not change during adsorption. As the phototransformation constantly changes the concentration of many different products in solution and as phototransformation is also influenced by the adsorption a kinetic examination is impossible.
Comment 3: The conclusion section should be more concise.
The conclusion has been reworked and specified.
Reviewer 2 Report
The manuscript by Fischer and co-workers describe the use of PVDF as an adsorbent during the photodegradation of a drug substance. The topic is of interest to the water treatment community. A polymer was used in the study but the polymer aspects of the work is non-existent. Therefore the manuscript might be more suitable for other MDPI journals such as Water. The manuscript is well-written but the discussions are somewhat lengthy. The presentation of the existing data must be improved and new experiments and data interpretations are needed to meet publication standards. There are several major issues that must be addressed before further consideration by the journal.
1, The threshold concentration for diclofenac is 0.1 μg L-1, and the usual concentrations found in contaminated water is 1 μg L-1, but the authors tested the efficiency of their method with 25 mg L-1. Such a high concentration is not justified. Why did the authors studied roughly 25,000–250,000 times higher concentrations than what is practically relevant. The feasibility and efficiency of the proposed method should be demonstrated at relevant concentrations.
2, The amount of both washing solutions (ethanol and water) should be precisely mentioned as ‘mL ethanol per gram (or cm2) of PVDF’ and ‘mL water per gram (or cm2) of PVDF’. Report the amount of water purified versus the amount of ethanol/water used for the PVDF regeneration. This will give an honest account for the actual water recovery, and the effectiveness of the method.
3, The PVDF adsorbent should not be called a membranes in this work as it was not used as a membrane. PVDF was a simple adsorbent and could have been a fiber, film or a bead. Membranes must act as a selective barrier between two phases. Moreover, the PVDF was not used in any filtration experiment but simply immersed inside an aqueous solution.
4, The intended application of the proposed methodology is water treatment, which is considered a large scale industrial application. The authors should discuss the scalability of the proposed methodology. How is the scale up of the ‘UV irradiation of water in the presence of PVDF in a petri dish’ envisaged?
5, The regeneration of PVDF was shown up to 3 attempts. Regeneration cycles are usually reported for 5-10 cycles to be able to see a trend. At least 5 cycles should be reported in Figure 7.
6, Adsorption isotherm and kinetics study should be included in the manuscript as the work is incomplete in its current form. These two studies are necessary for any adsorption based research to reveal the performance. Raw data shown in Figure 3 for instance is not sufficient in a research article.
7, Adsorption capacities need to be reported. Currently percentage values are given in the figures and the text and they have no real meaning and cannot be practically interpreted.
8, A final table in the manuscript should be added that compares the diclofenac removal efficiency compared to existing methods in the literature. A demonstration of advancement should be evident from the table. Adsorption capacities, water recovery, used agents/polymers/materials per purified water among other parameters should be reported.
Author Response
Dear Editior,
We appreciate the thorough examination of the manuscript by the reviewers, and we carefully revised the manuscript for obtaining now a better quality of the paper. The changes have been made using the "Track Changes" function in Microsoft Word. Therefore, I would like to resubmit our manuscript and hope we could eliminate all mistakes and misunderstandings according to the reviewer’s suggestions.
For easier review, we directly answered the reviewers comments (see below). Finally, I would like to thank the reviewers for the very helpful comments and work he/she put in our manuscript!
With best regards,
Agnes Schulze
The manuscript by Fischer and co-workers describe the use of PVDF as an adsorbent during the photodegradation of a drug substance. The topic is of interest to the water treatment community. A polymer was used in the study but the polymer aspects of the work is non-existent. Therefore the manuscript might be more suitable for other MDPI journals such as Water. The manuscript is well-written but the discussions are somewhat lengthy. The presentation of the existing data must be improved and new experiments and data interpretations are needed to meet publication standards. There are several major issues that must be addressed before further consideration by the journal.
Comment 1: The threshold concentration for diclofenac is 0.1 μg L-1, and the usual concentrations found in contaminated water is 1 μg L-1, but the authors tested the efficiency of their method with 25 mg L-1. Such a high concentration is not justified. Why did the authors studied roughly 25,000–250,000 times higher concentrations than what is practically relevant. The feasibility and efficiency of the proposed method should be demonstrated at relevant concentrations.
The goal of this study was to show the possibility to use a polymer (PVDF) as an adsorbent for products of UVA transformation of diclofenac to increase the removal of diclofenac from water. The intended field of application is of course contaminated water with diclofenac concentrations which are below the here tested concentration. Quantitative analytic determination of low concentration of diclofenac in water are extensive. HPLC is not possible, 1 µg L-1 is at the detection threshold. Evaluation via TOC measurement is also below the detection limit. Further on, the absorption at this concentration is too low to measure with UV-Vis absorption, which is a major point of this work, because we see the “inner” filter effect clearly. As mentioned above, the goal was just to show a new concept to combine adsorption and UVA transformation. To show this effect a concentration of 25 mg L-1 was chosen as the analytic methods (HPLC, TOC and UV-Vis) give reliable results.
Comment 2: The amount of both washing solutions (ethanol and water) should be precisely mentioned as ‘mL ethanol per gram (or cm2) of PVDF’ and ‘mL water per gram (or cm2) of PVDF’. Report the amount of water purified versus the amount of ethanol/water used for the PVDF regeneration. This will give an honest account for the actual water recovery, and the effectiveness of the method.
As mentioned above, the goal of this study was to show a new concept to combine adsorption and UVA phototransformation. The amount of washing solution was not optimized in any way; thus the same amount of washing solution was used as the volume with which it was treated (25 mL). The information about the volume of water and ethanol has been added in the experimental section.
Comment 3: The PVDF adsorbent should not be called a membranes in this work as it was not used as a membrane. PVDF was a simple adsorbent and could have been a fiber, film or a bead. Membranes must act as a selective barrier between two phases. Moreover, the PVDF was not used in any filtration experiment but simply immersed inside an aqueous solution.
The term “membrane” was eliminated from the manuscript. In the experimental section the term membrane was kept.
Comment 4: The intended application of the proposed methodology is water treatment, which is considered a large scale industrial application. The authors should discuss the scalability of the proposed methodology. How is the scale up of the ‘UV irradiation of water in the presence of PVDF in a petri dish’ envisaged?
Scale up is possible by transforming the technology into a continuous way. A solution of diclofenac in water is flowing by PVDF with simultaneous UVA irradiation. The following sentences in line 426-430 were added to emphasize this.
Line 426-430: The development of a continuous reactor is required to scale up the process. Further studies will be undertaken to study diclofenac removal in a continuous reactor where diclofenac in water is flowing over PVDF with simultaneous UVA irradiation. Different volume, concentration, flow velocities and surface area will be changed and tested.
Comment 5: The regeneration of PVDF was shown up to 3 attempts. Regeneration cycles are usually reported for 5-10 cycles to be able to see a trend. At least 5 cycles should be reported in Figure 7.
Experiments have been performed for 5 cycles and the section has been reworked based on the attempted results.
Comment 6: Adsorption isotherm and kinetics study should be included in the manuscript as the work is incomplete in its current form. These two studies are necessary for any adsorption based research to reveal the performance. Raw data shown in Figure 3 for instance is not sufficient in a research article.
Adsorption kinetics cannot be calculated as UVA phototransformation of diclofenac is taken place at the same time of adsorption of these products to PVDF. To calculate adsorption kinetics the amount of products in solution which adsorb on PVDF has to be known and should not change during adsorption. As the phototransformation constantly changes the concentration of many different products in solution and as phototransformation is also influenced by the adsorption a kinetic examination is impossible. The “inner” filter effect influence the transformation additionally.
Comment 7: Adsorption capacities need to be reported. Currently percentage values are given in the figures and the text and they have no real meaning and cannot be practically interpreted.
As discussed above – it is not possible to gain adsorption capacities as the concentration of the adsorbent is unknown, changing not only due to adsorption but also due to UVA reaction over time.
Comment 8: A final table in the manuscript should be added that compares the diclofenac removal efficiency compared to existing methods in the literature. A demonstration of advancement should be evident from the table. Adsorption capacities, water recovery, used agents/polymers/materials per purified water among other parameters should be reported.
As mentioned above – it is not possible to gain kinetics and adsorption capacities. Further on, the goal of this study was to show the possibility to use a polymer (PVDF) as an adsorbent for products of UVA transformation of diclofenac to increase the removal of diclofenac from water. The parameter of the experiment where chosen to see the factor of an adsorbent PVDF clearly in comparison to UVA irradiation without PVDF. The amount of adsorbent, used water, water for recovery tests are not optimized in any way. Therefore, the results of the presented manuscript are not qualified to be compared with other methods at this point. Further studies to optimize the parameters have to be undertaken to fairly compare the results.
Reviewer 3 Report
I read carefully the manuscript entitled ‘Enhanced Photodegradation of Diclofenac in the Presence of a Polyvinylidene Difluoride Membrane for Polymers. This research work addressed the usefulness of PVDF membrane for the removal of degraded products during the photodegradation of diclofenac. They have optimized the process parameters plus also checked the toxicity analysis. This research work is more comprehensive and ultimately fits and suitable to publish in Polymers. This manuscript is generally well written and clearly presented however still need to address many comments and require major revision before its acceptance.
- Title need to modify which can describe whole research work. Provide a nice graphical abstract representing the research work.
- In abstract authors should put the obtained result values. Add one or two sentences describing the importance of research work. Give the full form of PVDF in abstract.
- Authors should clarify whether this process is photodegradation or phototransformation throughout the manuscript and correct accordingly. Similarly Pl check throughout the manuscript whether degraded product or products.
- In the introduction section, write the novelty of the work and the problem statement clearly. From Ln no 82-97 these are repeated sentences of abstract should delete. Ln no 100-101 not understand clearly rewrite. The manuscript is also lacking some important recent review of literature for example photocatalytic degaradtion of dyes Journal of environmental management 223, 1086-1097, 2018. Last para of Introduction should describe the objectives of the research work.
- Materials and Methods section TOC measurement need to give detailed procedure. Toxicity assay information is difficult to understand give full details.
- Statistical analysis of the results should be provided in the materials and methods section. Its important for all experimental work Report these values in the results and discussion.
- Line no 243-244 author mentioned possible growth of microorganism is this really possible? How they checked. Pl clarify whether microbial or algae growth very confusing.
- After degradation of diclofenac which products have strong affinity towards PVDF and to improve the adsorption process what would be the future directions?
- Write the practical applications and future research perspectives of this work by adding new section before conclusions. Comparative table of results with the literature is recommended.
- Very surprisingly there is no any recent papers have been cited try to add some recent literature and from MDPI journals.
- The conclusion of the study is not discussed with the specific output obtained from the study, it could be modified with precise outcomes with a take home message.
Author Response
Dear Editior,
We appreciate the thorough examination of the manuscript by the reviewers, and we carefully revised the manuscript for obtaining now a better quality of the paper. The changes have been made using the "Track Changes" function in Microsoft Word. Therefore, I would like to resubmit our manuscript and hope we could eliminate all mistakes and misunderstandings according to the reviewer’s suggestions.
For easier review, we directly answered the reviewers comments (see below). Finally, I would like to thank the reviewers for the very helpful comments and work he/she put in our manuscript!
With best regards,
Agnes Schulze
I read carefully the manuscript entitled ‘Enhanced Photodegradation of Diclofenac in the Presence of a Polyvinylidene Difluoride Membrane for Polymers. This research work addressed the usefulness of PVDF membrane for the removal of degraded products during the photodegradation of diclofenac. They have optimized the process parameters plus also checked the toxicity analysis. This research work is more comprehensive and ultimately fits and suitable to publish in Polymers. This manuscript is generally well written and clearly presented however still need to address many comments and require major revision before its acceptance.
Comment 1: Title need to modify which can describe whole research work. Provide a nice graphical abstract representing the research work.
The title has been modified to “Enhanced Removal and Toxicity Decline of Diclofenac by Combining UVA Treatment and Adsorption of Photoproducts to Polyvinylidene Difluoride” and a graphical abstract has been added.
Comment 2: In abstract authors should put the obtained result values. Add one or two sentences describing the importance of research work. Give the full form of PVDF in abstract.
Result values have been added in line 21-25.
Line 21-25: At 30 min and 18 h of UVA treatment, removal of diclofenac and total organic carbon increases from 56% to 65% and 18% to 54%, respectively, when PVDF is present. The toxicity of a UVA treated (18 h) diclofenac solution doubles (from 5 to 10, expressed in toxicity units), while no toxicity was detectable when PVDF is present during UVA treatment (TU = 0).
In line 32 the importance of the work has been added.
Line 32-34: The combination of UVA irradiation of diclofenac with adsorption of photoproducts to PVDF is unique and opens up new possibilities to enhance removal of pollutants from water.
The full form of PVDF has been added in line 19.
Comment 3: Authors should clarify whether this process is photodegradation or phototransformation throughout the manuscript and correct accordingly. Similarly Pl check throughout the manuscript whether degraded product or products.
Authors changed photodegradation to phototransformation as degradation is defined as a reaction where a chemical substance is broken down to smaller molecules. Diclofenac is broken down but also dimerization is taking place to gain larger molecules. Also degraded products have been checked and changed accordingly.
Comment 4: In the introduction section, write the novelty of the work and the problem statement clearly. From Ln no 82-97 these are repeated sentences of abstract should delete. Ln no 100-101 not understand clearly rewrite. The manuscript is also lacking some important recent review of literature for example photocatalytic degaradtion of dyes Journal of environmental management 223, 1086-1097, 2018. Last para of Introduction should describe the objectives of the research work.
Part of the introduction has been reworked, literature added, line 82-97 deleted and line 100-101 rewriten.
The drawbacks and problems of AOPs have been added and a final sentence formulated to show the problem/objective clearly.
Line 63-71: The main drawbacks of photocatalysis are the high band energy of the semiconductor material, which are mostly not visible light active and show poor activity, agglomeration of used nanoparticles, the difficulty to separate the catalyst after photocatalysis from the aqueous media and the reusability of the catalyst [27]. Ozonation or degradation of pollutants via hydrogen peroxide are always accompanied by the addition of toxic and expensive oxidants. Also, AOPS generate several oxidation by-products with a high toxic potential [28]. Conventional treatment methods have many problems and drawbacks. Therefore, further studies to evaluate new strategies to remove pollutants like diclofenac from water are highly needed.
The novelty of the work has been added.
Line 95-103: In this work, an adsorbing material (hydrophobic polyvinylidene difluoride, PVDF) is introduced during UVA irradiation of diclofenac. Up to date PVDF was only applied as a support for photocatalysts [42,43], or MIPs (molecularly imprinted polymers nanoparticles) [44] which removed diclofenac from water. Here, PVDF adsorbs phototransformation products but not diclofenac itself [44]. The inhibiting products are drawn from the solution to reduce the “inner” filter effect and increase UVA transformation of diclofenac. So far, it is the first time, that UVA irradiation and adsorption of phototransformation products are combined to show an increased removal of a pollutant from water. This work demonstrates a new concept, which opens up new possibilities for pollutant elimination from water.
Comment 5: Materials and Methods section TOC measurement need to give detailed procedure. Toxicity assay information is difficult to understand give full details.
Detailed procedure has been added for TOC measurement in line 173.
Line 173-176: The total organic carbon (TOC) was quantified (triplicate) at a liquiTOC II (Elementar, Langenselbold, Germany) with high temperature oxidation between 850-900 °C and a wide range NDIR photometer. The concentration range was set to 20 ppm and calibration was performed prior to the tests. Samples were collected in glass vials at a volume of 25 ml and covered with tin foil.
Information on the toxicity assay has been added and reworked.
Line 189-205 : The toxicity tests on the marine luminescent bacteria Vibrio fischeri were performed according to DIN EN ISO 11348-1: 2009-05 by the Umweltanalytische Labor of the Institut für Siedlungswasserwirtschaft of RWTH Aachen with a LUMIStox using LUMIStherm System (Hach Lange GmbH, Düsseldorf, Germany). The test consisted of seven dilutions. The pH of the samples was corrected to 6.0 (SD = 0.2) before toxicity assessment tests using 0.1 M NaOH. According to the applied procedure, the self-produced (freshly grown) bacteria were reconstituted with reactivation solution, to provide a stock suspension of test organisms which was maintained at 15 °C and used to perform the test. The measurement of the bioluminescence was carried out at the start and after 30 min and the luminescence of the dilutions was compared with the negative control. A positive control was also included. Each dilution was tested in parallel. The data were automatically evaluated with the program LUMISsoft (Hach Lange GmbH, Düsseldorf, Germany) and the statistics program ToxRat (ToxRat Solutions GmbH, Alsdorf, Germany). The toxicity was expressed as toxicity units (TU) and GL levels. The TU was calculated by dividing 100 by the effective concentration causing 50% reduction of bioluminescence (TU = 100/EC50%). The GL value is the dilution level at which the inhibition is below 20% for the first time. All values below 20% inhibition are normal test fluctuations. Level 1 (80% volume fraction), level 4 (25% volume fraction), level 6 (16.66% volume fraction), level 8 (12.5% volume fraction).
Comment 6: Statistical analysis of the results should be provided in the materials and methods section. Its important for all experimental work Report these values in the results and discussion.
Information on the statistical analysis has been added in the experimental section (line 160-161).
Line 160-161: The standard deviation of the data was the basic statistical tool to give the amount of variation of the stated values. The confidence interval was estimated for the TU values (toxicity).
The standard deviation and confidence level has been added in the results and discussion section.
Comment: 7: Line no 243-244 author mentioned possible growth of microorganism is this really possible? How they checked. Pl clarify whether microbial or algae growth very confusing.
The experiments have been thoroughly repeated and analysis has been checked. Possible growth was not detected, thus the sentence was removed.
Comment 8: After degradation of diclofenac which products have strong affinity towards PVDF and to improve the adsorption process what would be the future directions?
We found that product X, has an affinity to PVDF, but as the analytical method used here (GC-MS) can not detect all products, we can not state how strong the affinity of product X is and if there are other products which adsorb to PVDF. Future directions would be to analyze all products and change the adsorbent accordingly. Either the surface of PVDF can be modified (change in hydrophobicity, change of surface charge, adding functional groups) or other polymeric materials can be used. A section has been added in the conclusion (line 422-425).
Line 422-425: This concept of increasing transformation by adsorbing phototransformation products which inhibit the reaction could be enhanced with other adsorbing polymers (e.g. PES, PTFE, PE) having different properties (e.g. surface charge, hydrophilicity, functional groups). As the polymer does not have to be irradiated itself, many different materials can be applied.
Comment 9: Write the practical applications and future research perspectives of this work by adding new section before conclusions. Comparative table of results with the literature is recommended.
A section was added (also due to the wish of the other reviewer) at the end of the conclusion section.
Line 425-430: This concept of increasing transformation by adsorbing phototransformation products which inhibit the reaction could be enhanced with other adsorbing polymers (e.g. PES, PTFE, PE) having different properties (e.g. surface charge, hydrophilicity, functional groups). As the polymer does not have to be irradiated itself, many different materials can be applied. The concept could be also adopted to other UVA phototransformable pollutants like sulfamethoxazole or ketoprofen. The development of a continuous reactor is required to scale up the process. Further studies will be undertaken to study diclofenac removal in a continuous reactor where diclofenac in water is flowing over PVDF with simultaneous UVA irradiation. Different volume, concentration, flow velocities and surface area will be changed and tested.
Comparison of the results at this stage of experimental work (petri-dish experiment, high concentration) is difficult. Also, no adsorption isotherms or kinetics could be gained as partially unknown phototransformation products evolve and the concentration changes due to two factors: UVA irradiation and adsorption.
Comment 10: Very surprisingly there is no any recent papers have been cited try to add some recent literature and from MDPI journals.
Recent literature from MDPI journal has been added.
Saeid, S.; Kråkström, M.; Tolvanen, P.; Kumar, N.; Eränen, K.; Mikkola, J.-P.; Kronberg, L.; Eklund, P.; Aho, A.; Palonen, H.; et al. Pt Modified Heterogeneous Catalysts Combined with Ozonation for the Removal of Diclofenac from Aqueous Solutions and the Fate of by-Products. Catalysts 2020, 10
Guedes-Alonso, R.; Montesdeoca-Esponda, S.; Pacheco-Juárez, J.; Sosa-Ferrera, Z.; Santana-Rodríguez, J.J. A Survey of the Presence of Pharmaceutical Residues in Wastewaters. Evaluation of Their Removal using Conventional and Natural Treatment Procedures. Molecules 2020, 25
Arguello-Pérez, M.Á.; Mendoza-Pérez, J.A.; Tintos-Gómez, A.; Ramírez-Ayala, E.; Godínez-Domínguez, E.; Silva-Bátiz, F.d.A. Ecotoxicological Analysis of Emerging Contaminants from Wastewater Discharges in the Coastal Zone of Cihuatlán (Jalisco, Mexico). Water 2019
Saratale, R.G.; Ghodake, G.S.; Shinde, S.K.; Cho, S.-K.; Saratale, G.D.; Pugazhendhi, A.; Bharagava, R.N. Photocatalytic activity of CuO/Cu(OH)2 nanostructures in the degradation of Reactive Green 19A and textile effluent, phytotoxicity studies and their biogenic properties (antibacterial and anticancer). Journal of Environmental Management 2018, 223, 1086–1097
Casillas, J.E.; Campa-Molina, J.; Tzompantzi, F.; Carbajal Arízaga, G.G.; López-Gaona, A.; Ulloa-Godínez, S.; Cano, M.E.; Barrera, A. Photocatalytic Degradation of Diclofenac Using Al2O3-Nd2O3 Binary Oxides Prepared by the Sol-Gel Method. Materials 2020, 13
Kudlek, E. Decomposition of Contaminants of Emerging Concern in Advanced Oxidation Processes. Water 2018, 10
Comment 11: The conclusion of the study is not discussed with the specific output obtained from the study, it could be modified with precise outcomes with a take home message.
The conclusion has been modified and a take a home message has been added.
Line 431-434: Combining UVA irradiation and adsorption of phototransformation products to a polymer could improve the elimination of pollutants from water. The concept is unique, improving different parameters like adsorbing polymer, UV irradiation and reactor design, could lead to a promising technique to remove UV phototransformable pollutants from water.
Round 2
Reviewer 2 Report
The manuscript has not improved enough to meet the scientific publication standards. The issue with the incorrect concentration range was simply ignored by saying that it is too difficult to work at low concentrations. If the authors chose to investigate a particular application then it makes no sense to work at 25,000–250,000 times higher concentrations than what is practically relevant. Te authors also replied that they did not bother to optimize their methods 'in any way'. Again, it is not acceptable. The adsorption isotherms and kinetics could have been studied as the adsorption can be deconvoluted from the UV induced degradation by simply not applying UV and performing the adsorption studies in a closed/amber environment.
The above comments refer to the replies to the previous comments # 1,2,7-8. These are crucial for the study to be complete and meet the minimum requirements of a scientific study. Although the work could have some merit if the proposed methodology was studied in depth, unfortunately in its current form the research is superficial and scientifically not sound.
Author Response
“The paper is well organized with interesting results that deserve to be published. However, authors need to take a look at the existing references. It is important to support the investigation approach used by comparing the results obtained with existing studies and reporting new insights."
Dear Mr. Li,
thank you again for taking your valuable time to let our manuscript evaluated again for publication in polymers. We have received the comment of an academic editor of polymers and are very thankful for the received advice. We added some new parts to compare our results with existing studies. Please see below the changed parts. Changes which were made can be also followed in the manuscript by the track change function.
Comment of academic editor:
“The paper is well organized with interesting results that deserve to be published. However, authors need to take a look at the existing references. It is important to support the investigation approach used by comparing the results obtained with existing studies and reporting new insights."
In “3.2. Transformation of diclofenac via UVA light in the presence of PVDF”: Line 264-275 following part has been added.
Overall, phototransformation depends on many parameters like molar absorption coefficient, quantum yield, intensity of emitted irradiation (incident photon flux), radiation wavelength, photoreactor design. Thus, direct comparison with the literature values is difficult. Diclofenac (30 and 10 mg L-1) is transformed by UVC irradiation (254 nm) within 9 min [52] and 6 min [53], respectively. Complete phototransformation of diclofenac (30 mg L-1) via UVA (365 nm) is reported to take 29 h [52]. UVA irradiation of diclofenac (0.5 mg L-1) with similar wavelengths (313 – 578 nm) results in lower transformation of 20% after 60 min [28] compared to over 90% in this study. Diclofenac is excited by light up to wavelengths of 320 nm with an absorption maximum at 278 nm (see Figure 3 (b)). Thus, the difference in phototransformation kinetics in literature to the here reached values is due to (amongst others) the applied irradiation wavelength. The TOC removal after complete diclofenac transformation is higher in this study compared to literature value of < 10% for UVA and UVC irradiation [52].
Line 283-286:
TOC removal via UVA irradiation is highly increased with PVDF present. Even strong UVC irradiation by Kovacic et al. [52] resulted in lower TOC removal of 10% after 100 min of irradiation compared to values of 26% and 40% after 60 min and 120 min in this study, respectively.
In the conclusion following sentences have been added:
Line 434-436:
TOC removal of diclofenac via UV irradiation without PVDF is found to be low in the literature (<10%), but can be raised by the addition of PVDF to over 26%.
Line 438-439:
As UVA degradation of diclofenac is incomplete, toxicity increases. The addition of PVDF enhances the removal and toxic phototransformation products are removed from the solution.
Line 449-454 last section of conclusion (words in yellow have been added):
Combining UVA irradiation and adsorption of phototransformation products to a polymer could improve the elimination of many other pollutants from water and decrease its toxicity. Toxic phototransformation products which are generated during UV irradiation of pollutants are removed via adsorption. The concept is unique, improving different parameters like adsorbing polymer, UV irradiation and reactor design, could lead to a promising technique to remove UV phototransformable pollutants and its transformation products from water.
Reviewer 3 Report
Authors have substantially revised the manuscript according to raised comments. The present form of the manuscript can be accepted for publication in Polymers.
Author Response

(The authors gave the same response as above.)
